# Molecular and clinicopathological differences between depressed and protruded T2 colorectal cancer

Kenichi Mochizuki[1,2☯], Shin-ei Kudo[1], Kazuki Kato[1,2☯], Koki Kudo[1], Yushi Ogawa[1], Yuta Kouyama[1], Yuki Takashina[1], Katsuro Ichimasa[1,3], Taro Tobo[4], Takeo Toshima[2], Yuichi Hisamatsu[2], Yusuke Yonemura[2], Takaaki Masuda[2], Hideyuki Miyachi[1], Fumio Ishida[1], Tetsuo Nemoto[5], Koshi Mimori[2]*

1 Digestive Disease Center, Showa University Northern Yokohama Hospital, Yokohama, Japan, 2 Department of Surgery, Kyushu University Beppu Hospital, Beppu, Japan, 3 Department of Gastroenterology and Hepatology, National University Hospital, Singapore, Singapore, 4 Department of Clinical Laboratory, Kyushu University Beppu Hospital, Beppu, Japan, 5 Department of Diagnostic Pathology, School of Medicine, Showa University, Yokohama Northern Hospital, Kanagawa, Japan

☯ These authors contributed equally to this work.
* mimori.koshi.791@m.kyushu-u.ac.jp

**Data Availability Statement:** All data files are available from the DRYAD database (https://doi.org/10.5061/dryad.6wwpzgn18).

## Abstract

### Background

Colorectal cancer (CRC) can be classified into four consensus molecular subtypes (CMS) according to genomic aberrations and gene expression profiles. CMS is expected to be useful in predicting prognosis and selecting chemotherapy regimens. However, there are still no reports on the relationship between the morphology and CMS.

### Methods

This retrospective study included 55 subjects with T2 CRC undergoing surgical resection, of whom 30 had the depressed type and 25 the protruded type. In the classification of the CMS, we first defined cases with deficient mismatch repair as CMS1. And then, CMS2/3 and CMS4 were classified using an online classifier developed by Trinh et al. The staining intensity of CDX2, HTR2B, FRMD6, ZEB1, and KER and the percentage contents of CDX2, FRMD6, and KER are input into the classifier to obtain automatic output classifying the specimen as CMS2/3 or CMS4.

### Results

According to the results yielded by the online classifier, of the 30 depressed-type cases, 15 (50%) were classified as CMS2/3 and 15 (50%) as CMS4. Of the 25 protruded-type cases, 3 (12%) were classified as CMS1 and 22 (88%) as CMS2/3. All of the T2 CRCs classified as CMS4 were depressed CRCs. More malignant pathological findings such as lymphatic invasion were associated with the depressed rather than protruded T2 CRC cases.

**Funding:** This work was supported by the Japan Society for the Promotion of Science KAKENHI. The grant numbers are JP19K09176(TM) and JP19H03715(KM). The URL of funder website is "https://kaken.nii.ac.jp/en/index/". The funders had no role in study design, data collection and analysis, decision to publish, or preparation of the manuscript.

**Competing interests:** The authors have declared that no competing interests exist.

## Conclusions

Depressed-type T2 CRC had a significant association with CMS4, showing more malignant pathological findings such as lymphatic invasion than the protruded-type, which could explain the reported association between CMS4 CRC and poor prognosis.

## Introduction

Colorectal cancer (CRC) is the fourth most diagnosed cancer globally and the second most common in terms of mortality. It is the third most commonly diagnosed malignancy in men and the second in women, with more than 1.8 million new cases and 880,000 deaths reported in 2018 [1]. In Japan, CRC is the third most common cancer in males after prostate and stomach cancers and the second most common in females after breast cancer [2]. In Japan, CRC-related death is the second most common cancer-related death after lung cancer [3]. Thus, comprehension of the development of advanced CRC is necessary for proper prevention and treatment.

In 2015, Guinney et al. reported four consensus molecular subtypes (CMS) of CRC according to gene expression [4]. Of these subtypes, CMS1 contains the majority of tumors with microsatellite instability and amplification of genes associated with immune infiltration. CMS2/3 tumors are characterized as epithelial. CMS4 tumors show elevated gene expression related to epithelial–mesenchymal transition (EMT), angiogenesis and TGF-β signaling and have the worst prognosis among the four subtypes. In addition to predicting prognosis, the CMS classification is expected to aid selection of the chemotherapy regimen [5–9]. Furthermore, immunohistochemistry (IHC)-based CMS of CRC, according to immunostaining of five oncogenic proteins (caudal type homeobox 2 (CDX2), FERM domain containing 6 (FRMD6), 5-hydroxytryptamine receptor 2B (HTR2B), zinc finger e-box binding homeobox 1 (ZEB1), and keratin (KER)), was reported by Trinh et al. in 2017. Previously, gene expression analysis was required for CMS classification, but this classifier has reduced the cost of, and facilitated, CMS classification without the need for gene expression analysis [10].

Among the morphological types of CRC, depressed-type CRC is more prone to submucosal and lymphatic invasion and has a higher risk of recurrence compared with the other morphological types [11, 12]. Next generation sequencing revealed higher expression of genes related to angiogenesis and EMT in depressed CRC in our previous study [11], suggesting that those specific traits could give depressed CRC the features of CMS4. On the other hand, few reports have focused on the morphology of T2 CRC with invasion limited to the muscularis propria [13]. In addition, no previous reports are available on depressed-type T2 CRC focusing on its CMS, although we sometimes encounter depressed T2 CRC cases with exacerbated malignant behavior after resection. Therefore, in the current study, we will evaluate the morphological characteristics of depressed T2 CRC according to the IHC-based CMS classification, as a conventional and robust methodology. The adenoma–carcinoma sequence and the de novo pathway, which are hypotheses of carcinogenic pathways in CRC, are reportedly associated with the protruded and depressed morphologies, respectively (Fig 1) [14]. The present analysis focused on the molecular characteristics of the protruded and depressed types.

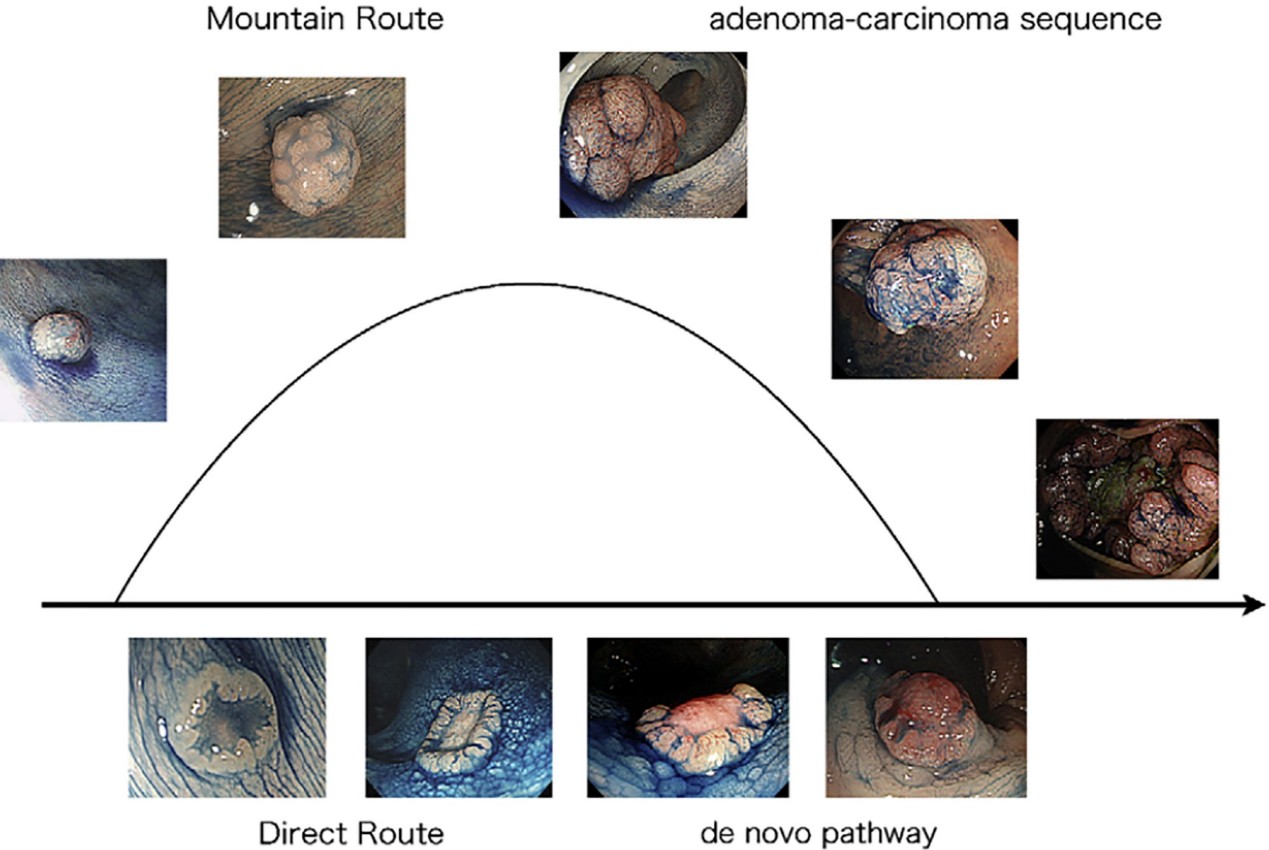

**Fig 1. The adenoma–carcinoma sequence and de novo theory.** The two main theories regarding the carcinogenic pathway of CRC. In the adenoma–carcinoma sequence, the lesions initially present as protruded and eventually become ulcerated. On the other hand, depressed lesions develop via the de novo pathway. Kudo et al. referred to these two pathways as the "mountain route" and "direct route", respectively.

## Methods

### Patients

Among patients with primary CRC who were surgically resected at Showa University Northern Yokohama Hospital between January 2010 and December 2013, 153 with stage T2 according to the TNM Classification of Malignant Tumors, 8th edition, of the Union for International Cancer Control were enrolled retrospectively. Patients who were diagnosed with Lynch syndrome (n = 3), ulcerative colitis (n = 6), or familial adenomatous polyposis (n = 6) were excluded. Patients lacking detailed clinical data were also excluded (n = 34). We did not include patients who received preoperative chemotherapy or radiotherapy. Tumor morphology was classified according to the Japanese Classification of Colorectal, Appendiceal, and Anal Carcinoma: the 3rd English edition [13]. In the present study, we focused on the protruded and depressed types, which are associated with the carcinogenic pathway, taking into account the classification of early-stage cancer and the classification of Mori et al. [15]. Of the stage T2 patients, 49 with other morphologies such as the flat type and ulcerative type were excluded. The remaining 55 patients were included in the study, of whom 30 had the depressed-type and 25 the protruded-type (Fig 2). The morphological type was retrospectively

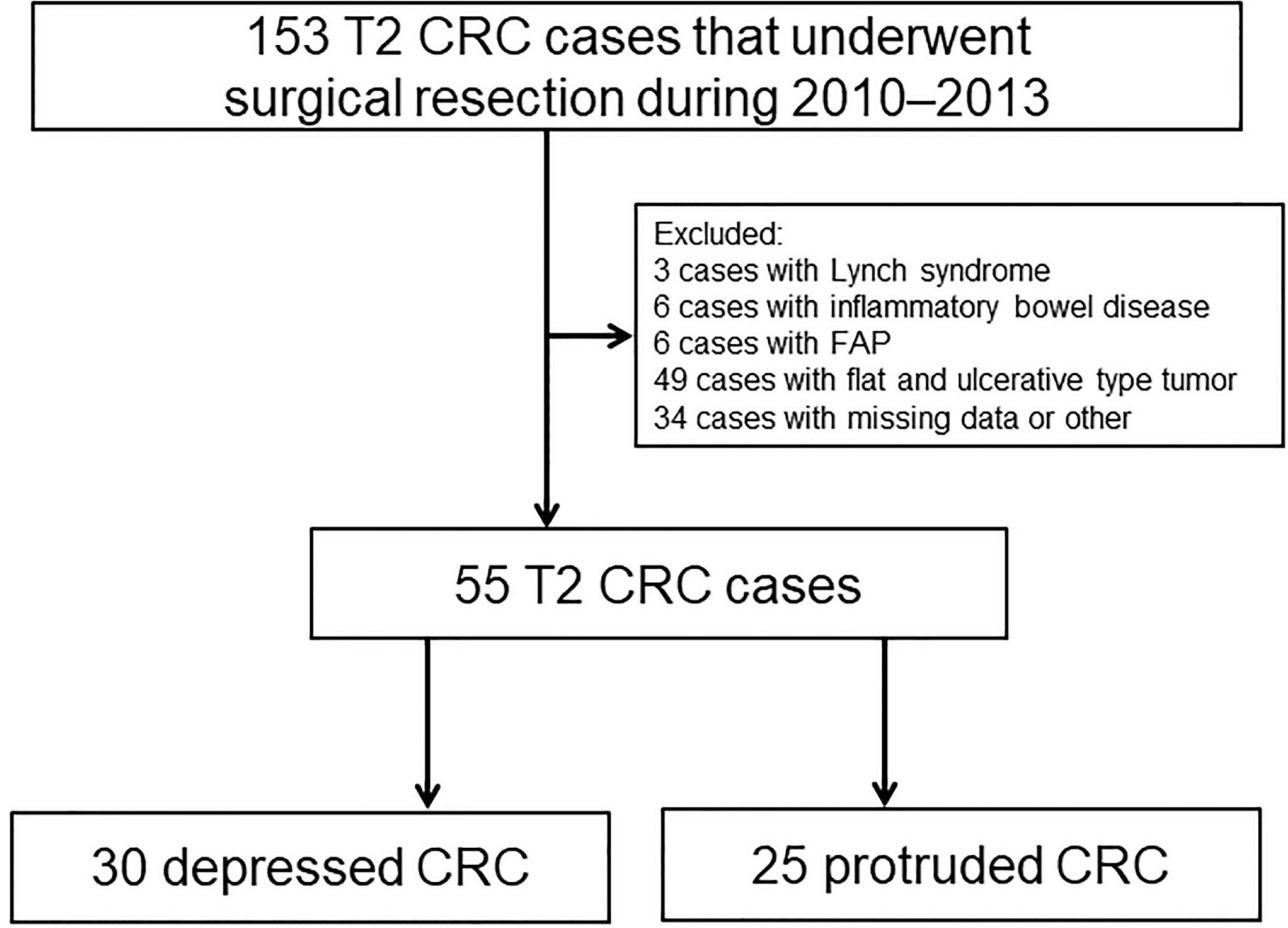

**Fig 2. Patient selection flow chart.** Of the 153 T2 colorectal cancer patients, 55 were included. 30 had the depressed-type and 25 the protruded-type.

determined by several endoscopic experts (K.M., K.K., Y.O., Y.K., H.M.). For reference, representative endoscopic images of both morphological types are shown in Fig 3. We determined their morphological type after indigo carmine staining. We analyzed their clinicopathological characteristics including patient age, sex, tumor size, morphology, histological grade, vascular invasion, lymphatic invasion, and lymph node metastasis. The ethical review committee of Kyushu University and Showa University Northern Yokohama Hospital approved the protocol of this study (approval numbers: 2021–200 and 21-057-A, respectively). Informed consent was obtained using an opt-out method by posting the study information on the relevant institutional website (https://beppu.kyushu-u.ac.jp/geka/wp2/wp-content/uploads/2021/07/ disclosure_20210415.pdf and https://www.showa-u.ac.jp/albums/abm.php?d=505&f= abm00031209.pdf&n=210715-1.pdf, respectively). This study was registered in the University Hospital Medical Network Clinical Trials Registry (UMIN000045702). The clinicopathological characteristics of the patients were collected from the electronic medical records of Showa University Northern Yokohama Hospital.

### Histological examination, mismatch repair (MMR) status, and IHC for classifying CMS

All resected specimens were retrieved and immediately fixed in 10% buffered formalin. The histological specimens were cut into parallel sections and stained with hematoxylin and eosin.

(a) (b)

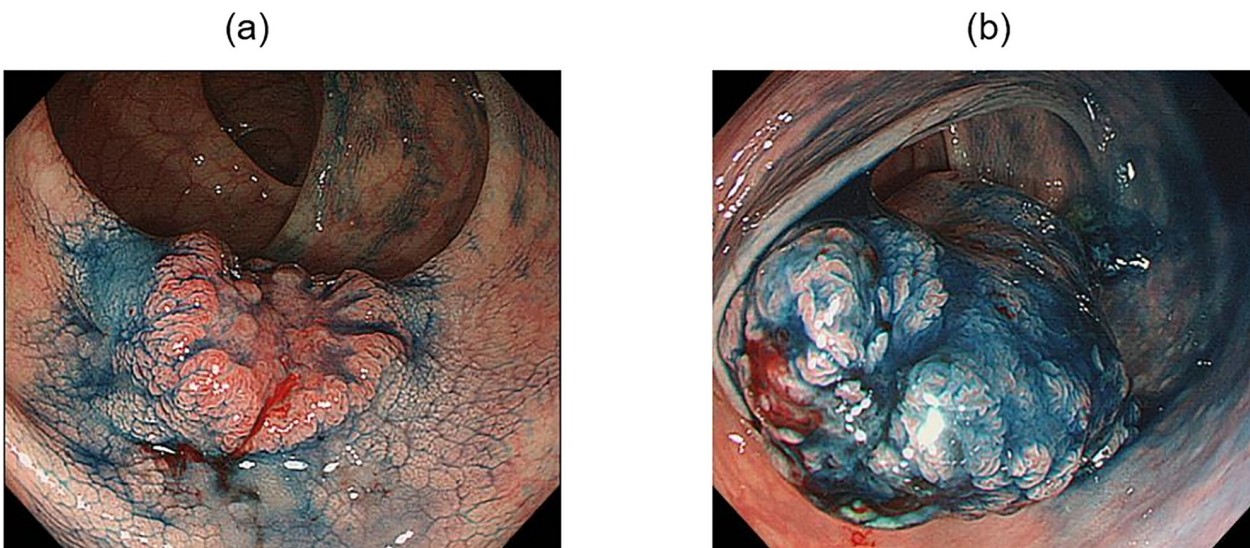

**Fig 3. Typical endoscopic images of each morphological type.** (a) Endoscopic image of a typical depressed-type colorectal cancer. A clear depressed area is seen after indigo carmine staining. (b) Endoscopic image of a typical protruded-type colorectal cancer.

Tumor size was measured after formalin fixation. All specimens were diagnosed based on the World Health Organization Classification of Tumors [16] and the current Japanese Society for Cancer of the Colon and Rectum guidelines [17]. The histological grade was classified according to the World Health Organization as well-differentiated adenocarcinoma, moderately differentiated adenocarcinoma, poorly differentiated adenocarcinoma, or mucinous carcinoma. In this study, the most abundant histology in the specimens was defined as the histological type of the lesion.

The MMR status of the patients was determined by evaluating the expression of MLH1, MSH2, MSH6, and PMS2 by IHC. The lack of expression of one or more of these proteins was defined as deficient MMR. CDX2, HTR2B, FRMD6, ZEB1, and KER were selected as markers for IHC for use with the CMS classifier, as reported by Trinh et al. These five markers were selected based on previous transcriptomic analyses. CDX2 is highly expressed in epithelial tumors and HTR2B in mesenchymal tumors. FRMD6 is a marker of goblet cells contained within mesenchymal tumors, and ZEB1 is a marker of EMT. KER, which is highly expressed in epithelial tumors, was used for normalization of the expression of the other markers. Immunostaining was performed using standard techniques. First, the obtained tissue sections were deparaffinized with xylene and a concentration-graded ethanol solution. After antigen activation in an autoclave at 121˚C for approximately 20 minutes, the sections were treated with 0.3% hydrogen peroxide at room temperature for 10 minutes to inhibit endogenous peroxidase activity. After blocking with bovine serum albumin, the dried slides were incubated with the primary antibody and stored in a refrigerator at 4˚C overnight; the primary antibodies used were anti-CDX2 (1:200; Novus Biologicals; NB100-2136), anti-HTR2B (1:75; Sigma-Aldrich; HPA012867), anti-FRMD6 (1:500; Sigma-Aldrich; HPA001297), anti-ZEB1 (1:500; Sigma; HPA027524), and anti-cytokeratin (AE1/AE3; 1:500; Thermo Scientific; 41-9003-82). The next day, EnVision™ HRP, Rabbit/Mouse (K5007; Dako, Glostrup, Denmark) was added dropwise, followed by chromogenic treatment with DAB. The sections were contrast-stained with hematoxylin, and the immunostaining reaction was evaluated. To use the published classifier, the immunostaining results were determined as follows. For CDX2, FRMD6, and KER,

both the staining intensity (low, middle, or high intensity) and the stained area were evaluated. For CDX2, which is expressed in the nucleus, nuclei with positive and negative staining were counted, and the percentage of positive nuclei was calculated as the percentage of area. Because FRMD6 and KER are expressed in the cytoplasm, the area of FRMD6 or KER staining was determined objectively. For HTR2B, only the staining intensity was evaluated (low, middle, or high intensity). ZEB1 immunostaining was classified only as positive or negative; the stained area was evaluated in the maximum field of view (×400), and the average percentage in five fields of view was calculated. Immunostaining evaluations for all slides were conducted by K.M. under the guidance of an experienced pathologist (T.T.).

## CMS classifier based on IHC

In the classification of the CMS, we first defined cases with deficient MMR as CMS1. CMS2/3 and CMS4 were classified using an online classifier (https://crcclassifier.shinyapps.io/appTesting/, accessed on 18 November 2021) developed by Trinh et al. The staining intensity of CDX2, HTR2B, FRMD6, ZEB1, and KER and the percentage contents of CDX2, FRMD6, and KER are input into the classifier to obtain automatic output classifying the specimen as CMS2/3 or CMS4.

## Statistical analysis

Nominal and ordinal variables are expressed as the number and percentage of patients. Continuous variables are reported as the median with the interquartile range. Continuous variables were compared using the Wilcoxon rank-sum test, and dichotomous variables were compared using the Chi-square test or Fisher's exact test as appropriate. All statistical analyses were performed using R v4.0.5 (The R Foundation for Statistical Computing). All P values are two-sided, and $P < 0.05$ is considered statistically significant.

# Results

## Representative immunostaining images

Representative images of the CDX2, FRMD6, KER, HTR2B, and ZEB1 immunostaining results are shown in Fig 4, and the staining results are summarized in Table 1. In this study, only a high staining intensity was considered positive. When comparing the depressed and protruded morphologies, the area of CDX2 staining and the intensity and area of KER staining were significantly lower in the depressed than protruded lesions.

## Clinicopathological characteristics according to each CMS

The CMS classification of each morphological type, determined using the online classifier, and the clinicopathological characteristics according to CMS classification are summarized in Table 2. All of the cases classified as CMS4 were the depressed-type, and the incidence of the depressed-type was significantly higher in the CMS4 than CMS1 and CMS2/3 groups. In addition, the CMS4 group exhibited a significantly higher rate of lymphatic invasion compared with the CMS2/3 group. The CMS4 group tended to have a higher rate of lymph node metastasis compared with the other two groups, but the difference was not statistically significant. There were no obvious differences in the other characteristics between the CMS2/3 and CMS4 groups. Because only three cases were classified as CMS1, statistical differences compared with CMS2/3 and CMS4 were difficult to determine.

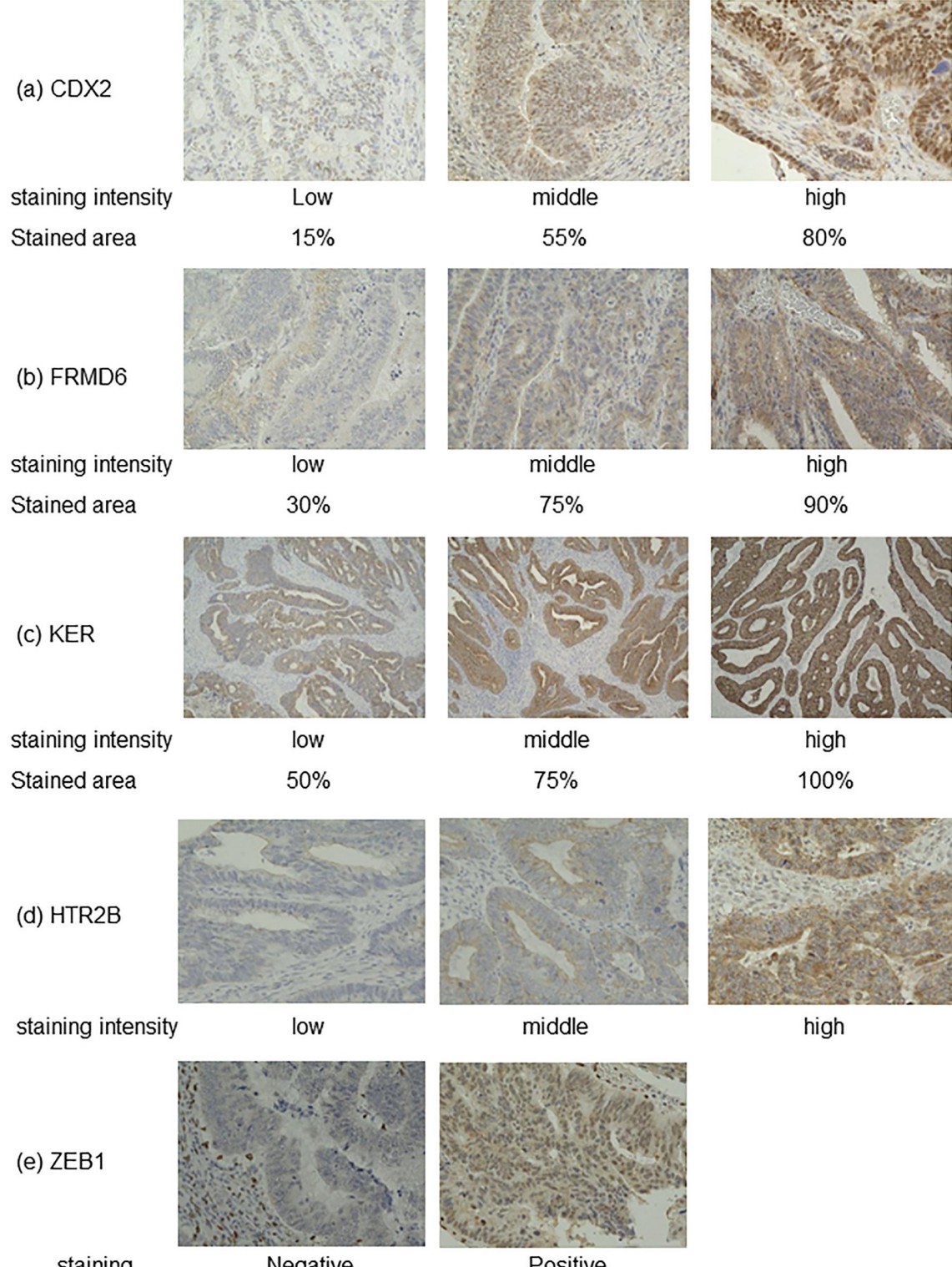

**Fig 4. Typical immunostaining images for each marker.** (a) CDX2, (b) FRMD6, (c) KER, (d) HTR2B, and (e) ZEB1.

**Table 1. Immunostaining results and consensus molecular subtype (CMS) classification according to morphological type (depressed/protruded).**

|  | Depressed (n = 30) | Protruded (n = 22) | *P*-value |
|---|---|---|---|
| CDX2 (positive) | 23 (76.7%) | 17 (77.3%) | 1.00 |
| **CDX2 (area)** | 55 (50–65) | 70 (65–85) | **<0.001** |
| FRMD6 (positive) | 7 (23.3%) | 3 (13.6%) | 0.488 |
| FRMD6 (area) | 85 (70–90) | 72.5 (58.8–93.8) | 0.355 |
| **KER** (positive) | 5 (16.7%) | 11 (50.0%) | **0.015** |
| **KER (area)** | 60 (51.3–68.8) | 70 (61.3–75) | **0.002** |
| HTR2B (positive) | 5 (16.7%) | 8 (36.4%) | 0.121 |
| ZEB1 (positive) | 7 (23.3%) | 5 (22.7%) | 1.00 |

The percentage of positive staining and stained area for each marker are shown. In this study, only a high staining intensity was considered positive. The table shows the number (%) of positive cases. The stained area is defined as the median (inter quartile range) percentage of stained area in the field of view.

## Clinicopathological characteristics of the patients with the depressed versus protruded type

In this study, 55 patients surgically treated for T2 CRC were enrolled. The median age of the enrolled patients was 64 years (range, 25–83 years), and 29 (52.7%) were male. The other clinicopathological characteristics are shown in Table 3. The depressed-type lesions had a significantly smaller diameter (23 vs. 30 mm, $P = 0.002$) and a higher rate of lymphatic invasion (70.0% vs. 40.0%, $P = 0.032$) and tended to have higher rates of vascular invasion (76.7% vs. 52.0%, $P = 0.087$) and lymph node metastasis (46.7% vs. 28.0%, $P = 0.177$) compared with the protruded-type lesions. There were no obvious differences in age, sex, or histological grade according to morphological type. Of the 30 depressed-type cases, 0 were classified as CMS1, 15 (50%) as CMS2/3, and 15 (50%) as CMS4, whereas among the 25 protruded-type cases, 3 (12%) were classified as CMS 1, 22 (88%) as CMS2/3, and 0 as CMS4. Depressed-type lesions is associated with CMS4 ($P < 0.001$).

**Table 2. Clinicopathological characteristics according to each consensus molecular subtype (CMS).**

|  | CMS1 (n = 3) | CMS2/3 (n = 37) | CMS4 (n = 15) | *P*-value |
|---|---|---|---|---|
| **Morphology (depressed)** | 0 (0.0%) | 15 (40.5%) | 15 (100%) | **< 0.001** |
| Age (years) | 65 (63.5–73.5) | 64 (55–70) | 61 (48.5–69) | 0.550 |
| Sex (male) | 1 (33.3%) | 22 (59.5%) | 6 (40.0%) | 0.350 |
| Location (rectum) | 0 (0.0%) | 12 (32.4%) | 4 (26.7%) | 0.479 |
| Tumor size (mm) | 26 (24–29) | 26 (21–34) | 23 (20–25) | 0.059 |
| Histological grade (Por or Muc[a]) | 0 (0.0%) | 1 (2.7%) | 0 (0.0%) | 0.781 |
| Vascular invasion (+) | 0 (0.0%) | 27 (73.0%) | 9 (60.0%) | 0.150 |
| **Lymphatic invasion (+)** | 2 (66.7%) | 16 (43.2%) | 13 (86.7%) | **0.016** |
| Lymph node metastasis (+) | 1 (33.3%) | 11 (29.7%) | 9 (60.0%) | 0.124 |

Clinicopathological characteristics by CMS are shown. Age and tumor size are expressed as the median (interquartile range). All other variables are expressed as the number of patients. The numbers in parentheses indicate the percentage of applicable cases.

[a] Por or Muc: poorly differentiated adenocarcinoma or mucinous carcinoma.

**Table 3. Clinicopathological characteristics of patients according to morphological type (depressed/protruded).**

| | Depressed (n = 30) | Protruded (n = 25) | Total (n = 55) | *P*-value |
|---|---|---|---|---|
| Age (years) | 64.5 (51.3–69.8) | 63 (55–70) | 64 (53.5–70) | 0.793 |
| Sex (male) | 15 (50.0%) | 14 (56.0%) | 29 (52.7%) | 0.788 |
| Location (rectum) | 10 (33.3%) | 6 (24.0%) | 16 (29.1%) | 0.556 |
| **Tumor size (mm)** | 23 (20–25.8) | 30 (23–38) | 25 (20.5–30) | **0.002** |
| Histological grade (Por or Muc[a]) | 0 (0.0%) | 1 (4.0%) | 1 (1.8%) | 0.455 |
| Vascular invasion (+) | 23 (76.7%) | 13 (52.0%) | 36 (65.5%) | 0.087 |
| **Lymphatic invasion (+)** | 21 (70.0%) | 10 (40.0%) | 31 (56.4%) | **0.032** |
| Lymph node metastasis (+) | 14 (46.7%) | 7 (28.0%) | 21 (38.2%) | 0.177 |
| CMS1 | 0 (0%) | 3 (12%) | 3 (5.5%) | **<0.001** |
| CMS2/3 | 15 (50%) | 22 (88%) | 37 (67.3%) | |
| CMS4 | 15 (50%) | 0 (0%) | 15 (27.4%) | |

Clinicopathological characteristics by morphology are shown. Age and tumor size are expressed as the median (interquartile range). All other variables are expressed as the number (%) of patients.

[a] Por or Muc: poorly differentiated adenocarcinoma or mucinous carcinoma.

## Discussion

In this study, we compared 30 cases of depressed CRC and 25 cases of protruded CRC among surgically resected T2 CRC specimens. The CDX2 and KER staining areas were smaller, and the rate of CMS4 was higher, among the depressed than protruded CRCs. The depressed CRCs had a smaller size and a higher rate of lymphatic invasion compared with the protruded CRCs.

CDX2 is expressed during the formation of the intestinal tract, playing an important role in its development and maintenance [18]. It has also been reported that CDX2 expression is inversely correlated with the malignancy of CRC [19] and inhibits EMT and metastasis [20, 21] by regulating Snail expression and β-catenin stabilization via PI3K/Akt/GSK-3β signaling [22]. The low expression of CDX2 in depressed CRC may support the high malignancy of depressed CRC, which is consistent with the finding of elevated expression of EMT-related genes in depressed CRC by Kudo et al. [11]. KER is highly expressed in tumors of epithelial origin, differentiating them from mesenchymal tumors. The low expression of KER in depressed CRC supports the notion that EMT occurs in these tumors [23]. On this basis, it has been reported that keratin loss due to phosphorylation is associated with EMT [24]. The obtained immunostaining results were used to output the CMS classification using the online classifier. This online classifier is available to the public online, and by inputting the numerical values of the immunostaining area and intensity, the CMS classification is output automatically. As a result, all cases classified as CMS4 were depressed CRC. The transcriptomic profiles of the CMS4 CRC cases comprised malignancy-related genes, such as accelerating EMT, immune tolerance, and a worse prognosis compared with CMS1, 2, and 3 [4]. The immunostaining status was concordant with the characteristics of CMS4 CRC. The concordance of the molecular characteristics of CMS4 with those of depressed CRC, and the classification of all CMS4 cases as the depressed type, demonstrated the strong association between CMS4 and depressed CRC.

Krijn et al. reported that the proportion of CMS4 among T1 CRC lesions, which have an approximately 10% rate of lymph node metastasis [25, 26], is very small (1.8%) [27]. They attributed this to the rapid progression of CMS4 tumors and the difficulty in detecting CMS4 at the T1 stage [27]. It has been reported that depressed CRC is difficult to detect at earlier

stages because of the small size, weak color change, and stealth shape without obvious protuberance [14, 28], which may be additional evidence supporting that CMS4 CRC originates from depressed CRC. Considering the more malignant traits of CMS4, early detection and resection of depressed CRC might be required to prevent systemic advancement and to improve clinical outcome. Kouyama et al. compared the prognosis of T1 CRC patients who underwent endoscopic resection versus surgical resection and found that those who underwent endoscopic resection had a significantly poorer prognosis [29]. Therefore, depressed CRC may have a strong ability to invade deep intestinal layers and rapidly progress to systemic disease. Therefore, we need to conduct strict follow-up monitoring of depressed CRC patients even after curative endoscopic resection. CMS4 is associated with a poor prognosis and is often not detected until the advanced stages (III-IV) [4]. Considering the high pathological grade of the depressed type, depressed CRC with CMS4 usually requires chemotherapy. Although the association between the CMS and long-term outcomes could not be demonstrated due to the small number of cases in this study, we plan to accumulate more cases prospectively as a practical application of this study to determine whether morphology and CMS classification are candidate predictors of the need for adjuvant chemotherapy.

In this study, T2 CRC was used for analysis because it is easier to clarify its characteristics such as lymph node metastasis and distant metastasis according to morphological type compared with T1 CRC. Mori et al. evaluated T2 CRC according to morphology and reported a smaller size and higher rate of positive lymphatic invasion in depressed T2 CRC. They found no significant differences in the rates of lymph node metastasis and distant metastasis, which they suggested was due to their low rates even in T2 CRC, emphasizing the need for accumulation of more cases [15]. The same finding of a significantly higher rate of positive lymphatic invasion in depressed T2 CRC was observed in the present study. There was also a tendency for a higher rate of positive lymph node metastasis in the depressed-type, although the difference was not significant.

There are several limitations of this study. First, the patients were from a single center, potentially introducing regional or institutional selection bias. Because there was only a single evaluator of the immunostaining results, observer bias was also possible. However, as a single-center study, there was the advantage that the treatment procedures, pathology and immunostaining decisions, and surveillance methods were consistent across all patients. Second, we applied IHC-based CMS to classify depressed T2 CRC according to microsatellite instability status and immunostaining using an online classifier. However, this method could not distinguish between CMS2 and CMS3. Therefore, the classifier yielded three categories: CMS1, CMS2/3, and CMS4. Third, this study was conducted retrospectively and was not randomized, and thus it was subjected to selection bias. Fourth, we excluded flat type and ulcerative type cases of CRC. In this study, we focused on the two hypotheses of carcinogenic pathways in CRC: protruded type by the adenoma-carcinoma sequence and depressed type by the de novo pathway. We speculate that lesions with endoscopically depressed localization rarely have an adenoma component in the pathology [11] and take different carcinogenic pathway. The flat type and ulcerative type are comprised of a mixture of both protruded and depressing elements and a mixture of tumors that have followed many carcinogenic pathways, such as those originating from serrated lesions [30] or granular-type laterally spreading tumors (G-LST) [31]. Therefore, it was appropriate first to clarify the two representative carcinogenic pathways of CRC, the adenoma-carcinoma sequence and the de novo pathway, and clarification of the flat and ulcerative type was considered an issue to be addressed later. We excluded flat and ulcerated type T2 CRCs. However, there were 49 cases of flat and ulcerated type, which is not negligible and may have a large selection bias. Serrated pathway carcinogenesis has been reported to include MSI carcinomas [30], and G-LST has been reported to have a High prevalence of

CpG island methylator phenotype-high [31]. These are consistent with the characteristics of CMS1. Perhaps the morphology excluded in this study includes CMS1 cases, which may have represented only a minority of cases in this study.

In conclusion, such higher proportion of depressed-type T2 CRCs among CMS4 cases could explain the reported association between CMS4 CRC and poor prognosis.

## Acknowledgments

We thank M. Kasagi, S. Sakuma, T. Fukuda, N. Mishima, and T. Kawano for their technical assistance.

## Author Contributions

**Conceptualization:** Shin-ei Kudo, Koshi Mimori.

**Formal analysis:** Kenichi Mochizuki.

**Funding acquisition:** Takaaki Masuda, Koshi Mimori.

**Investigation:** Kenichi Mochizuki, Kazuki Kato, Taro Tobo, Fumio Ishida, Tetsuo Nemoto.

**Methodology:** Kazuki Kato, Koki Kudo, Yushi Ogawa, Yuta Kouyama, Yuki Takashina, Katsuro Ichimasa, Takeo Toshima, Yuichi Hisamatsu, Yusuke Yonemura, Takaaki Masuda, Hideyuki Miyachi.

**Project administration:** Kenichi Mochizuki.

**Supervision:** Koshi Mimori.

**Visualization:** Kenichi Mochizuki.

**Writing – original draft:** Kenichi Mochizuki.

**Writing – review & editing:** Shin-ei Kudo, Yushi Ogawa, Yuta Kouyama, Yuki Takashina, Katsuro Ichimasa, Hideyuki Miyachi, Fumio Ishida, Tetsuo Nemoto, Koshi Mimori.

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
