## [Decision Letter · Decision Letter 0]

14 Apr 2022

PONE-D-22-01736Molecular and clinicopathological features of depressed T2 colorectal cancerPLOS ONE

Dear Dr. Mimori,

Thank you for submitting your manuscript to PLOS ONE. After careful consideration, we feel that it has merit but does not fully meet PLOS ONE’s publication criteria as it currently stands. Therefore, we invite you to submit a revised version of the manuscript that addresses the points raised during the review process.

Specifically, you should provide valid reasons why flat and ulcerative lesions were excluded from the analysis.

We look forward to receiving your revised manuscript.

Kind regards,

Peh Yean Cheah, Ph.D.

Academic Editor

PLOS ONE

Journal Requirements:

Reviewers' comments:

Reviewer's Responses to Questions

**Comments to the Author**

1. Is the manuscript technically sound, and do the data support the conclusions?

Reviewer #1: No

Reviewer #2: Yes

2. Has the statistical analysis been performed appropriately and rigorously? 

Reviewer #1: Yes

Reviewer #2: Yes

3. Have the authors made all data underlying the findings in their manuscript fully available?

Reviewer #1: Yes

Reviewer #2: Yes

4. Is the manuscript presented in an intelligible fashion and written in standard English?

Reviewer #1: Yes

Reviewer #2: Yes

5. Review Comments to the Author

Reviewer #1: Mochizuki et al. analyzed 30 depressed type and 25 protruded type T2 colorectal cancers by immunohistochemistry-based CMS classification. CMS4 tumors were exclusive to depressed type and malignant pathological findings were more commonly observed in depressed type tumors compared with protruded type tumors.

I do not understand why the authors excluded flat type and ulcerative type lesions from the analysis. The comparison between depressed and protruded types is clearly insufficient to determine the clinicopathological features of depressed type lesions among T2 colorectal cancers. It is possible that CMS4 and malignant pathological findings are significantly less common in protruded type lesions among T2 tumors. Thus, the authors’ conclusion that depressed-type T2 colorectal cancers are significantly associated with CMS4 is not justified by the presented data.

Protein symbols should not be italicized.

Reviewer #2: Colorectal cancer (CRC) is gradually increasing and major cause of cancer death all over the world. Recent studies have clarified the molecular subtypes of CRC according to gene expression. However, the relationship between molecular profile and morphological types of CRC remains unclear. The authors focused on morphological types, depressed-type T2 CRC and showed significant association with CMS4, which have the worst prognosis among the four CMS subtypes. The findings in this study are very intriguing. However, there were several critical points to be corrected to meet the criteria of this journal.

Major Query

1. In this study, the authors focused on just depressed and protruded types of the four gross morphological classifications of T2 CRC. Why did the authors select these types? Also, why these 2 types were distinguished by molecular profiling? Please discuss and address about it in Introduction or Discussion section from clinical or molecular point of view.

2. The authors showed the relationship between expression of CDX2, KER and malignant grade of CRC. It is likely that both genes are associated with EMT pathway. Please discuss about molecular mechanism in detail. Also, please cite some papers regarding the association between both genes and EMT.

3. The authors showed no relationship between tumor location (rectum or not) and each CMS. How about is sidedness of colon cancer? Recently, the impact of sidedness of CRC on tumor immunity has been reported.

4. The authors described that depressed CRC has malignant behavior, requiring careful surveillance. How do the authors plan to make use of finding obtained from this study into clinical practice? For example, can depressed-type T2 CRC with CMS4 be candidate factor for the indication for adjuvant chemotherapy? Please describe about clinical perspective of this study.

Specific comments

1. In introduction section, the authors described about epidemiology of CRC. More worldwide information is needed, not just focusing on Japanese data.

2. We can find some typographical errors and different font size (e.g line 227 and 300). Please correct them.

3. Who determined the morphological type? Some experts of endoscopist probably did, please specify it in method section because it sometimes depends on endoscopist.

4. We miss the data about long-term outcome in this study. The reviewers think it weakness of this study. Please address about it in Discussion section.

6. PLOS authors have the option to publish the peer review history of their article (what does this mean?). If published, this will include your full peer review and any attached files.

Reviewer #1: No

Reviewer #2: No

---

## [Author Response · Author response to Decision Letter 0]

30 May 2022

Dear Editors, 

We thank the reviewers for their careful assessment of our manuscript and their useful comments. In response to the Reviewers’ comments, we have carefully revised the manuscript point-by-point. 

We hope that our manuscript is now suitable for publication in the International Journal of PLOS ONE and look forward to hearing from you at your earliest convenience.

Yours sincerely, 

Our responses to the Reviewers’ reports are as follows:

Response to the Reviewer

Thank you for your careful review of our paper. We have responded to each of your points below.

Reviewers' comments:

Reviewer #1: 

Major comments:

Mochizuki et al. analyzed 30 depressed types and 25 protruded type T2 colorectal cancers by immunohistochemistry-based CMS classification. CMS4 tumors were exclusive to the depressed type, and malignant pathological findings were more commonly observed in depressed type tumors compared with protruded type tumors.

I do not understand why the authors excluded flat type and ulcerative type lesions from the analysis. The comparison between depressed and protruded types is insufficient to determine the clinicopathological features of depressed type lesions among T2 colorectal cancers. CMS4 and malignant pathological findings may be significantly less common in protruded type lesions among T2 tumors. Thus, the authors’ conclusion that depressed-type T2 colorectal cancers are associated considerably with CMS4 is not justified by the presented data. 

Response: Thank you for your helpful comment. This study excluded flat and ulcerative types and focused our analysis on protruded and depressed types. The adenoma-carcinoma sequence and the de novo pathway, which are the representative carcinogenic pathway of CRC, have been reported to have protruded and depressed morphologies, respectively. To investigate the molecular biological characteristics of these two types, the present analysis focused on the protruded and depressed types.[1]. Attached are figures (Fig. 1) on two representative CRC carcinogenesis pathways reported by Kudo et al. As you pointed out, there is a hidden possibility that CMS4 and malignant pathological findings are less common in the protruded type. However, we believe that all CMS4 colorectal cancers were depressed-type colorectal cancers is very impactful data. We want to add flat or ulcerative lesions to our analysis as future work. We have made changes to the Introduction based on your suggestions. We have changed the description of the adoption of morphology as follows:

“The adenoma–carcinoma sequence and the de novo pathway, which are representative carcinogenic pathways of CRC, are reportedly associated with the protruded and depressed morphologies, respectively (Fig. 1). The present analysis focused on the molecular characteristics of the protruded and depressed types[14].” (Page 5, line 83)

Minor comments:

Protein symbols should not be italicized.

Response: Thank you for your comment. We corrected all italicization of proteins.

Reviewer #2: 

Major comments:

1. In this study, the authors focused on just depressed and protruded types of the four gross morphological classifications of T2 CRC. Why did the authors select these types? Also, why these 2 types were distinguished by molecular profiling? Please discuss and address about it in Introduction or Discussion section from clinical or molecular point of view.

Response: Thank you for your useful comment. In this study, our primary objective was to elucidate the molecular biological characteristics of depressed type colorectal cancer, which is reported to have a high malignant potential, and we referred to the research methods of the pioneering Kudo et al. The adenoma-carcinoma sequence and the de novo pathway, which are representative carcinogenic pathway of CRC, have been reported to have protruded and depressed morphologies, respectively. In order to investigate the molecular biological characteristics of these two types, the present analysis focused on the protruded and depressed types[1]. As you pointed out, there is a hidden possibility that CMS4 and malignant pathological findings are less common in the protruded type. However, we believe that all CMS4 colorectal cancers were depressed-type colorectal cancers is very impactful data. We want to add flat or ulcerative lesions to our analysis as future work. We have made changes to the Introduction based on your suggestions. We have changed the description of the adoption of morphology as follows:

“The adenoma–carcinoma sequence and the de novo pathway, which are representative carcinogenic pathways of CRC, are reportedly associated with the protruded and depressed morphologies, respectively (Fig. 1). The present analysis focused on the molecular characteristics of the protruded and depressed types [14].” (Page 5, line 83)

We also believe that molecular profiling has demarcated these two types as follows. First, Kudo et al. reported that depressed CRC is characterized by higher expression of genes related to angiogenesis and EMT. This is consistent with the characteristics of CMS4, and in this study, all of the patients classified as CMS4 were depressed type. Based on these findings, we hypothesized that depressed type CRC is more strongly associated with CMS4 compared to the protruded type. We have added the description of the molecular profiling of morphology as follows:

“The concordance of the molecular characteristics of CMS4 with those of depressed CRC, and the classification of all CMS4 cases as the depressed type, demonstrated the strong association between CMS4 and depressed CRC.” (Page 15, line 275)

2. The authors showed the relationship between expression of CDX2, KER and malignant grade of CRC. It is likely that both genes are associated with EMT pathway. Please discuss about molecular mechanism in detail. Also, please cite some papers regarding the association between both genes and EMT.

Response: Thank you for your very important point. Yu et al. reported that CDX2 inhibits EMT and metastasis of CRC by regulation of Snail expression and β-catenin stabilization through PI3K/Akt/GSK-3β signaling. Kim et al. also reported that keratin loss due to phosphorylation is associated with EMT. Based on the above, we have added the description of CDX2 and EMT as follows:

“It has also been reported that CDX2 expression is inversely correlated with the malignancy of CRC [19] and inhibits EMT and metastasis[20, 21] by regulating Snail expression and β-catenin stabilization via PI3K/Akt/GSK-3β signaling [22].” (Page 14, line 259)

We have added two references reporting on the association between CDX2 and EMT to page 14, line 261, as follows:

20. Liu H, Yan R, Liang L, Zhang H, Xiang J, Liu L, et al. The role of CDX2 in renal tubular lesions during diabetic kidney disease. Aging (Albany NY). 2021; 13(5): 6782-6803. https://doi.org/10.18632/aging.202537. PMID: 33621200.

21. Zhang JF, Qu LS, Qian XF, Xia BL, Mao ZB, Chen WC. Nuclear transcription factor CDX2 inhibits gastric cancercell growth and reverses epithelialtomesenchymal transition in vitro and in vivo. Mol Med Rep. 2015; 12(4): 5231-5238. https://doi.org/10.3892/mmr.2015.4114. PMID: 26238762.

On the other hand, Kim et al. reported that keratin loss due to phosphorylation is associated with EMT. Based on this, we have added the description of KER and EMT and a reference as follows:

“On this basis, it has been reported that keratin loss due to phosphorylation is associated with EMT [24].” (Page 15, line 267)

24. Kim HJ, Choi WJ, Lee CH. Phosphorylation and Reorganization of Keratin Networks: Implications for Carcinogenesis and Epithelial-Mesenchymal Transition. Biomol Ther (Seoul). 2015; 23(4): 301-312. https://doi.org/10.4062/biomolther.2015.032. PMID: 26157545 

3. The authors showed no relationship between each CMS's tumor location (rectum or not). How about is sidedness of colon cancer? Recently, the impact of sidedness of CRC on tumor immunity has been reported.

Response: Thank you for your critical comment. We did an additional analysis of the right or left side of the colon and whether it was rectal or not. The results were as follows, with no significant differences. This may be due to the small sample size. Since the relationship between tumor location and immunity is an essential point, we would like to study it again in the future with a more significant number of cases.

4. The authors described depressed CRC as malignant behavior requiring careful surveillance. How do the authors plan to use findings obtained from this study in clinical practice? For example, can depressed-type T2 CRC with CMS4 be a candidate factor for the indication for adjuvant chemotherapy? Please describe the clinical perspective of this study.

Response: Thank you for your helpful comment. CMS4 has a poor prognosis and is also often at an advanced stage (III-IV)[2]. Considering the high pathologic grade of the depressed type, depressed CRC with CMS4 is likely to require chemotherapy. Based on the results of this study alone, it is not possible to discuss chemotherapy and its effect on chemotherapy due to the small number of cases. In the future, we would like to collect prospective cases and study depressed colorectal cancer, CMS4, and the effects of chemotherapy. We have added the description of a future challenge as follows:

“CMS4 is associated with a poor prognosis and is often not detected until the advanced stages (III-IV)[4]. Considering the high pathological grade of the depressed type, depressed CRC with CMS4 usually requires chemotherapy. Although the association between the CMS and long-term outcomes could not be demonstrated due to the small number of cases in this study, we plan to accumulate more cases prospectively as a practical application of this study to determine whether morphology and CMS classification are candidate predictors of the need for adjuvant chemotherapy.” (Page 16, line 293)

Minor comments:

1. In the introduction section, the authors described the epidemiology of CRC. More worldwide information is needed, not just focusing on Japanese data. 

Response: Thank you for your helpful comment. We have added the description of the epidemiology of CRC as follows:

“It is the third most commonly diagnosed malignancy in men and the second in women, with more than 1.8 million new cases and 880,000 deaths reported in 2018 [1].” (Page4, line 51)

2. We can find some typographical errors and different font size (e.g line 227 and 300). Please correct them.

Response: Thank you for pointing this out. We have corrected the font.

3. Who determined the morphological type? Some experts of endoscopists probably did; please specify it in the method section because it sometimes depends on the endoscopist.

Response: Thank you for your helpful comment. We have added the description of the method of determining the morphological type as follows:

“The morphological type was retrospectively determined by several endoscopic experts (K.M., K.K., Y.O., Y.K., H.M.).” (Page6, line 111)

4. We miss the data about long-term outcomes in this study. The reviewers think it is a weakness of this study. Please address it in the Discussion section.

Response: Thank you for your helpful comment. Long-term outcomes were analyzed, but the results did not differ due to the small number of cases. Figure B and C of the analysis results are attached. We have added the description of the discussion as follows:

“Although the association between the CMS and long-term outcomes could not be demonstrated due to the small number of cases in this study, we plan to accumulate more cases prospectively as a practical application of this study to determine whether morphology and CMS classification are candidate predictors of the need for adjuvant chemotherapy.” (Page16, line 295)

1. Kudo SE, Takemura O, Ohtsuka K. Flat and depressed types of early colorectal cancers: from East to West. Gastrointest Endosc Clin N Am. 2008; 18(3): 581-593, xi. https://doi.org/10.1016/j.giec.2008.05.013. PMID: 18674705.

2. Guinney J, Dienstmann R, Wang X, de Reynies A, Schlicker A, Soneson C, et al. The consensus molecular subtypes of colorectal cancer. Nat Med. 2015; 21(11): 1350-1356. https://doi.org/10.1038/nm.3967. PMID: 26457759.

---

## [Decision Letter · Decision Letter 1]

23 Jun 2022

PONE-D-22-01736R1Molecular and clinicopathological features of depressed T2 colorectal cancerPLOS ONE

Dear Dr. Mimori,

Thank you for submitting your manuscript to PLOS ONE. After careful consideration, we feel that it has merit but does not fully meet PLOS ONE’s publication criteria as it currently stands. Therefore, we invite you to submit a revised version of the manuscript that addresses the points raised during the review process.

Please respond to reviewer 1's comments. Specifically,Change the title to better reflect the study cohort;Elaborate on the limitation of excluding the flat- and ulcerative-T2 lesions in the Discussion section since these two type of lesions contributed almost equal number as the protruded- and depressed-T2 colorectal cancers of your cohort (49 vs 55).Please submit your revised manuscript by Aug 07 2022 11:59PM. If you will need more time than this to complete your revisions, please reply to this message or contact the journal office at plosone@plos.org. Please include the following items when submitting your revised manuscript:A rebuttal letter that responds to each point raised by the academic editor and reviewer(s). You should upload this letter as a separate file labeled 'Response to Reviewers'.A marked-up copy of your manuscript that highlights changes made to the original version. You should upload this as a separate file labeled 'Revised Manuscript with Track Changes'.An unmarked version of your revised paper without tracked changes. You should upload this as a separate file labeled 'Manuscript'.

We look forward to receiving your revised manuscript.

Kind regards,

Peh Yean Cheah, Ph.D.

Academic Editor

PLOS ONE

Reviewers' comments:

Reviewer's Responses to Questions

**Comments to the Author**

1. If the authors have adequately addressed your comments raised in a previous round of review and you feel that this manuscript is now acceptable for publication, you may indicate that here to bypass the “Comments to the Author” section, enter your conflict of interest statement in the “Confidential to Editor” section, and submit your "Accept" recommendation.

Reviewer #1: (No Response)

Reviewer #2: All comments have been addressed

2. Is the manuscript technically sound, and do the data support the conclusions?

Reviewer #1: No

Reviewer #2: Yes

3. Has the statistical analysis been performed appropriately and rigorously? 

Reviewer #1: Yes

Reviewer #2: Yes

4. Have the authors made all data underlying the findings in their manuscript fully available?

Reviewer #1: Yes

Reviewer #2: Yes

5. Is the manuscript presented in an intelligible fashion and written in standard English?

Reviewer #1: Yes

Reviewer #2: Yes

6. Review Comments to the Author

Reviewer #1: This study analyzes immunohistochemistry-based consensus molecular subtype classification in depressed- and protruded-type T2 colorectal cancers and indicate that depressed-type T2 CRCs are more likely to show the CMS4 phenotype and malignant pathological findings.

If the authors want to elucidate the “molecular and clinicopathological features of depressed T2 colorectal cancer”, they should also analyze flat- and ulcerative-type lesions. Otherwise, the authors should extensively revise the manuscript, including the title, to clearly indicate that this study analyzed the differences between depressed- and protruded-type T2 colorectal cancer.

The authors added Figure 1, suggesting that protruded and depressed lesions are related to two major tumorigenic pathways. However, what are the tumorigenic processes of flat and ulcerative lesions, which constituted 49 cases in this cohort?

Reviewer #2: The authors sufficiently responded to all queries. The reviewer think that this study meets all applicable standards for the ethics of experimentation and research integrity.

7. PLOS authors have the option to publish the peer review history of their article (what does this mean?). If published, this will include your full peer review and any attached files.

Reviewer #1: No

Reviewer #2: No

---

## [Author Response · Author response to Decision Letter 1]

29 Jul 2022

Response to the Reviewer

Thank you for your careful review of our paper. We have responded to each of your points below.

Reviewer's comments:

Reviewer #1: 

This study analyzes immunohistochemistry-based consensus molecular subtype classification in depressed- and protruded-type T2 colorectal cancers and indicate that depressed-type T2 CRCs are more likely to show the CMS4 phenotype and malignant pathological findings.

If the authors want to elucidate the “molecular and clinicopathological features of depressed T2 colorectal cancer”, they should also analyze flat- and ulcerative-type lesions. Otherwise, the authors should extensively revise the manuscript, including the title, to clearly indicate that this study analyzed the differences between depressed- and protruded-type T2 colorectal cancer.

The authors added Figure 1, suggesting that protruded and depressed lesions are related to two major tumorigenic pathways. However, what are the tumorigenic processes of flat and ulcerative lesions, which constituted 49 cases in this cohort?

Response : Thank you for your perceptive comment. We changed the title to “Molecular and clinicopathological differences between depressed and protruded T2 colorectal cancer.” As you have concerned in the original manuscript, we have excluded findings related to the flat and ulcerated types, which should be discussed as the major limitation in this paper. Therefore, we have added a paragraph to discuss it as follows:

“Fourth, we excluded flat type and ulcerative type cases of CRC. In this study, we focused on the two hypotheses of carcinogenic pathways in CRC: protruded type by the adenoma-carcinoma sequence and depressed type by the de novo pathway. We speculate that lesions with endoscopically depressed localization rarely have an adenoma component in the pathology[11] and take their carcinogenic pathway. The flat type and ulcerative type are comprised of a mixture of both protruded and depressing elements and a mixture of tumors that have followed many carcinogenic pathways, such as those originating from serrated lesions[30] or granular-type laterally spreading tumors (G-LST)[31]. Therefore, it was appropriate first to clarify the two representative carcinogenic pathways of CRC, the adenoma-carcinoma sequence and the de novo pathway, and clarification of the flat and ulcerative type was considered an issue to be addressed later. We excluded flat and ulcerated type T2 CRCs. However, there were 49 cases of flat and ulcerated type, which is not negligible and may have a large selection bias. Serrated pathway carcinogenesis has been reported to include MSI carcinomas[30], and G-LST has been reported to have a High prevalence of CpG island methylator phenotype-high[31]. These are consistent with the characteristics of CMS1. Perhaps the morphology excluded in this study includes CMS1 cases, which may have represented only a minority of cases in this study.” (Page 16, line 319)

11. Kudo SE, Kouyama Y, Ogawa Y, Ichimasa K, Hamada T, Kato K, et al. Depressed Colorectal Cancer: A New Paradigm in Early Colorectal Cancer. Clin Transl Gastroenterol. 2020; 11(12): e00269. https://doi.org/10.14309/ctg.0000000000000269. PMID: 33512809.

30. O'Brien MJ, Zhao Q, Yang S. Colorectal serrated pathway cancers and precursors. Histopathology. 2015; 66(1): 49-65. https://doi.org/10.1111/his.12564. PMID: 25263173.

31. Hiraoka S, Kato J, Tatsukawa M, Harada K, Fujita H, Morikawa T, et al. Laterally spreading type of colorectal adenoma exhibits a unique methylation phenotype and K-ras mutations. Gastroenterology. 2006; 131(2): 379-389. https://doi.org/10.1053/j.gastro.2006.04.027. PMID: 16890591.

---

## [Editor Report · Decision Letter 2]

11 Aug 2022

Molecular and clinicopathological differences between depressed and protruded T2 colorectal cancer

PONE-D-22-01736R2

Dear Dr. Mimori,

We’re pleased to inform you that your manuscript has been judged scientifically suitable for publication and will be formally accepted for publication once it meets all outstanding technical requirements.

Kind regards,

Peh Yean Cheah, Ph.D.

Academic Editor

PLOS ONE

Additional Editor Comments (optional):

Please change this sentence in Abstract (line 43) to: More malignant pathological findings such as lymphatic invasion were associated with the depressed rather than protruded T2 CRC cases.Please change this sentence in Discussion (line 323) to: We speculate that lesions with endoscopically depressed localization rarely have an adenoma component in the pathology[11] and take different carcinogenic pathway.
---

## [Editor Report · Acceptance letter]

2 Sep 2022

PONE-D-22-01736R2 

Molecular and clinicopathological differences between depressed and protruded T2 colorectal cancer 

Dear Dr. Mimori:

I'm pleased to inform you that your manuscript has been deemed suitable for publication in PLOS ONE. Congratulations! Your manuscript is now with our production department. 

Kind regards, 

on behalf of

Dr. Peh Yean Cheah 

Academic Editor

PLOS ONE